# Screening the Presence of Non-Typhoidal *Salmonella* in Different Animal Systems and the Assessment of Antimicrobial Resistance

**DOI:** 10.3390/ani11061532

**Published:** 2021-05-24

**Authors:** Dácil Rivera, Kasim Allel, Fernando Dueñas, Rodolfo Tardone, Paula Soza, Christopher Hamilton-West, Andrea I. Moreno-Switt

**Affiliations:** 1Facultad de Ciencias de la Vida, Universidad Andres Bello, Republica 440, Santiago 8320000, Chile; dacil.rivera@unab.cl (D.R.); ferdu26@gmail.com (F.D.); rodolfo.vet@gmail.com (R.T.); paula.soza.ossandon@gmail.com (P.S.); 2Millennium Initiative for Collaborative Research on Bacterial Resistance (MICROB-R), Santiago 7550000, Chile; k.allel@ucl.ac.uk; 3Department of Disease Control, Faculty of Infectious & Tropical Diseases, London School of Hygiene & Tropical Medicine, London WC1E 7HT, UK; 4Antimicrobial Resistance Centre, London School of Hygiene & Tropical Medicine, London WC1E 7HT, UK; 5Departamento de Medicina Preventiva, Facultad de Ciencias Veterinarias y Pecuarias, Universidad de Chile, Santa Rosa 11735, La Pintana, Santiago 8820000, Chile; christopher.hamilton@veterinaria.uchile.cl; 6Escuela de Medicina Veterinaria, Pontificia Universidad Católica de Chile, Santiago 7810000, Chile

**Keywords:** non-typhoidal *Salmonella*, screening of *Salmonella*, *Salmonella* serogroup D, multidrug-resistant, antimicrobial-resistant, Chile

## Abstract

**Simple Summary:**

In this study, for the first time in Chile, we compared resistance profiles of *Salmonella* strains isolated from 4047 samples from domestic and wild animals. A total of 106 *Salmonella* strains (2.61%) were isolated, and their serogroups were characterized and tested for susceptibility to 16 different antimicrobials. This study reports 47 antimicrobial-resistant (AMR) *Salmonella* strains (44.3% of total strains). Of the 47, 28 corresponded to single-drug resistance (26.4%) and 19 to multidrug resistance (17.9%). The association between AMR and a subset of independent variables was evaluated using multivariate logistic models. Interestingly, *S.* Enteritidis was highly persistent in animal production systems; however, we report that serogroup D strains were 18 times less likely to be resistant to at least one antimicrobial agent than the most common serogroup (serogroup B). The antimicrobials presenting the greatest contributions to AMR were ampicillin, streptomycin and tetracycline.

**Abstract:**

*Salmonella* is a major bacterial foodborne pathogen that causes the majority of worldwide food-related outbreaks and hospitalizations. Salmonellosis outbreaks can be caused by multidrug-resistant (MDR) strains, emphasizing the importance of maintaining public health and safer food production. Nevertheless, the drivers of MDR *Salmonella* serovars have remained poorly understood. In this study, we compare the resistance profiles of *Salmonella* strains isolated from 4047 samples from domestic and wild animals in Chile. A total of 106 *Salmonella* strains (2.61%) are isolated, and their serogroups are characterized and tested for susceptibility to 16 different antimicrobials. The association between antimicrobial resistance (AMR) and a subset of independent variables is evaluated using multivariate logistic models. Our results show that 47 antimicrobial-resistant strains were found (44.3% of the total strains). Of the 47, 28 correspond to single-drug resistance (SDR = 26.4%) and 19 are MDR (17.9%). *S.* Enteritidis is highly persistent in animal production systems; however, we report that serogroup D strains are 18 times less likely to be resistant to at least one antimicrobial agent than the most common serogroup (serogroup B). The antimicrobials presenting the greatest contributions to AMR are ampicillin, streptomycin and tetracycline. Additionally, equines and industrial swine are more likely to acquire *Salmonella* strains with AMR. This study reports antimicrobial-susceptible and resistant *Salmonella* in Chile by expanding the extant literature on the potential variables affecting antimicrobial-resistant *Salmonella*.

## 1. Introduction

The global non-typhoidal salmonellosis burden in 2010 was estimated at 93.8 million cases and 155,000 deaths per annum, of which more than two million were accounted for in the region of the Americas [1]. The causative agent belongs to *Salmonella*’s genus, which has two species: *enterica and bongori* [2]. *S. enterica* contains six subspecies: *enterica*, *salamae, arizonae, diarizonae, houtenae* and *indica* [2]. According to the White–Kaufmann–Le Minor scheme, *Salmonella* has been classified into more than 2600 serovars [3]. This scheme is based on the somatic or O-antigen antigenic reactions to determine the *Salmonella* serogroup and further reactions of the flagellar antiserum for H1 and H2-antigens [4]. These antigenic reactions are used in conjunction with each other to classify the antigenic formula and, consequently, to determine *Salmonella* serovars [5,6]. Forty-six variants of the O-antigens are contained in the scheme [6]; however, only a few serogroups are the most frequently reported *Salmonella* serovars in humans and animals [7]. Frequent *Salmonella* serovars have been fundamentally found in serogroup D (i.e., Enteritidis, Dublin and Javiana), serogroup B (i.e., Typhimurium and Heidelberg), serogroup C1 (i.e., Infantis and Montevideo) and serogroup C2–C3 (i.e., Kentucky and Newport) [6].

Antimicrobial resistance (AMR) in *Salmonella* is a leading worldwide concern. The World Health Organization (WHO) has classified fluoroquinolone-resistant *Salmonella* as a high-priority target for new drug development [8]. Several outbreaks attributed to AMR *Salmonella* have been reported globally [9,10,11]. Surveillance systems in developed countries have shown variability in the current trends of *Salmonella* AMR due to their divergent tendencies when referring to the antimicrobial agent provoking resistance and the serovar type [12]. For instance, data from the National Antimicrobial Resistance Monitoring System of Enteric Bacteria (NARMS) from the United States showed that specific serovars (i.e., 4, 5, 12:i:-, Typhimurium, Newport and Heidelberg) were resistant to at least three antimicrobial classes [13]. This is commonly known as the presence of multidrug-resistant (MDR) strains. Similarly, serovars isolated in Europe exhibited varying prevalence of AMR [12], with *S.* Enteritidis more susceptible to different antimicrobial groups tested in up to 84.7% of examined isolates [12]. Likewise, *S.* Enteritidis tested at the US NARMS presented low AMR levels, accounting only for 0–7.7% of resistant strains [13].

In Chile, the presence of *Salmonella* strains has been reported in different sources including water used for irrigation [14], backyard flocks [15], wild animals [16,17,18,19] and chicken eggs [18]. One study depicted a prevalence of MDR *Salmonella* of 13% in 35 strains obtained from irrigation water [14]. Additionally, a recent article reported one strain of *S.* Infantis in a wild owl at a Chilean rehabilitation center, which was not only MDR *Salmonella* but also an extended-spectrum ß-lactamase producer [16]. These reports have raised the importance of studying environments to better understand the distribution of *Salmonella* and the mechanisms by which it acquires resistance to different antimicrobials. This study aims to compare the prevalence of *Salmonella* AMR by exploring how it differs between serogroup types, sampling sources and different animal populations (e.g., domestic and wild animals) in Chile [16].

## 2. Materials and Methods

### 2.1. Study Sites and Sample Collection

We selected different study sites to represent a wide diversity of animals and environments where wild animals and livestock are found in Chile. Samples were classified into eight categories to represent a broader spectrum of characteristics, such as family and animal types, and geographic locations within the country (Table 1). Fourth of the categories were classified as closed systems where animals have restricted movement, limited contact with other animal species and are feed by humans. The closed system category is composed of food production animals from eight industrial dairy farms (160 samples) [20] and 10 swine farms (182 samples), domestic animals from an equine veterinary hospital (545 samples) [21] and wild animals from three wildlife rehabilitation centers (405 samples) [19]. Samples were also obtained from animals living free-range outdoors, which were potentially being fed with unsafe remains of food, such as human waste [22]. The open-system category comprised samples obtained from 13 free-range dairy farms (260 samples), 329 backyard chicken farms (2188 samples), four sites of wetland birds (271 samples) and five other backyard animal sites (36 samples) (Table 1).

### 2.2. Salmonella Isolation

A total of 4047 fecal samples were collected from the sites described above between 2013 and 2017 for isolation of *Salmonella* spp. Two sample types were collected: (i) sterile containers from fresh cows, swine, horses and wild animal feces that were deposited in the environment, and (ii) samples directly obtained from animals (horses, wildlife and birds) (Table 1). For the latter, rectal or cloacal samples were extracted using Cary Blair transport media (Copan Italia Spa, Brescia, Italy). All samples were collected under sterile conditions and transported to the laboratory at 4 °C for further processing.

The microbiology isolation method used in this study has been previously described [21]. Samples were cultured in buffered peptone water (Beckton-Dickinson, Franklin Lakes, NJ, USA) at 37 °C for 24 h. One-hundred microliters were transferred into Rappaport Vassiliadis (RV) media (Beckton-Dickinson) supplemented with novobiocin (20 mg/mL), and 1 mL was transferred into tetrathionate (TT) (Beckton-Dickinson) supplemented with iodine. Consequently, these samples were incubated at 42 °C for 24 h. Finally, a 100 uL aliquot of each selective enrichment broth was streaked into an XLT-4 agar plate (Beckton-Dickinson) and incubated at 37 °C for additional 24 h. Four presumptive *Salmonella* colonies were selected from each plate and transferred into a non-selective enrichment media tryptic soy agar (TSA) (Beckton-Dickinson). Subsequently, they were confirmed as *S. enterica* strains via polymerase chain reaction (PCR) of the *invA* gene using previously described primers [23]. Confirmed isolates were stored at −80 °C with 20% glycerol concentration, and confirmed as *S. enterica* strains via polymerase chain reaction (PCR) of the *invA* gene using previously described primers [23].

### 2.3. Serogroup Characterization and Antimicrobial Susceptibility

A previously described molecular method for predicting the *Salmonella* serogroup was used in this study to determine the serogroups of *Salmonella* strains [7]. DNA extraction was conducted using a DNeasy Blood and Tissues kit (QIAGEN; Hilden, Germany). DNA was quantified, and its quality was tested using a 250/280 ratio in a MaestroNano Spectrophotometer (Maestro-gen, Taiwan). DNA was then adjusted to a concentration of 25 ng/uL. Multiplex PCR was conducted to identify the serogroup of each isolate [7]. The scheme detected the most common serogroups; B, C1, C2–C3 and D. Isolates that could not be classified were reported as not determined (ND) (Appendix A).

Antimicrobial resistance was determined by disk diffusion as previously reported [24]. Strains were cultured overnight in tryptic soy broth (TSB), from Beckton-Dickinson, at 37 °C. Upon incubation, a McFarland Standard of 0.5 in a DEN-1B Densitometer (Biosan, Latvia) was used to adjust the bacterial concentration. Mueller-Hinton Agar (Beckton-Dickinson) was used to test the susceptibility to amikacin (30 µg), amoxicillin/clavulanate (30 µg), ampicillin (10 µg), cefoxitin (30 µg), ceftriaxone (30 µg), ciprofloxacin (5 µg), chloramphenicol (30 µg), streptomycin (10 µg), gentamicin (10 µg), kanamycin (30 µg), trimethoprim/sulfamethoxazole (25 µg) and tetracycline (30 µg). The Clinical Laboratory Standard Institute, section on Veterinary Microbiology Standards Documents, was employed to classify susceptibility [25], using *Escherichia coli* ATCC 25,922 as control. For analytical purposes, AMR was defined using a dummy variable indicating resistance to at least one antibiotic. AMR included strains with single-drug resistance (SDR) to one or two antimicrobials, while those with resistance to three or more antimicrobials were MDR [26].

### 2.4. Map of the Isolation Sites

*Salmonella* strains with AMR and those that were MDR were georeferenced using a geographical positioning system (Gpsmap62s, Garmin, Olathe, Kansas). Subsequently, their spatial distribution was mapped using color codes and risk maps on ArcGIS 10 software (Esri, Redlands, CA, USA) based on the coordinates from each isolation site.

### 2.5. Statistical Analyses

We employed three analyses throughout the study to explore the presence of AMR and MDR *Salmonella* strains. (i) Firstly, we examined *Salmonella*’s presence and serogroups by employing descriptive statistics and exploratory analyses based on screening features of our sample. (ii) Secondly, we computed a multivariate hierarchical analysis to examine different *Salmonella* clusters with AMR, drawing on antimicrobial resistance and susceptibility profiles. (iii) Thirdly, two different regression models were employed to look at the association between *Salmonella* with AMR and our independent variables: serogroup type (B, C1, C2–C3, D, E1, N/D) and animal category (industrial swine, wetland birds, equine veterinary animals, free-range dairy animals and backyard chickens).

(ii) Multivariate hierarchical analysis. These analyses used agglomerative hierarchical clustering algorithms to evaluate how different resistance profiles were built upon. We used the Euclidean distance as the clustering method. Data were grouped into different rows and according to their standardized values in respect to their serogroup classification (B, C1, C2–C3, E1, ND), animal and system types (bird, chicken, cow, horse, small mammals, swine, reptile; closed and free-range) and antibiotic susceptibility (to amikacin, amoxicillin, ampicillin, cefoxitin, ceftriaxone, ciprofloxacin, chloramphenicol, streptomycin, gentamycin, kanamycin, trimethoprim/sulfamethoxazole and tetracycline). The results of the model are displayed in a dendrogram plot for data interpretation and visualization. This analysis was carried out using Infostat Software, version 2017 (https://www.infostat.com.ar/, accessed on 10 March 2021).

(iii) Regression models. First, a multivariate logistic model was used to understand how different biological and animal characteristics were associated with *Salmonella* AMR for the total number of samples collected (*n* = 4047). The biological and animal characteristics used as independent variables included animal categories (swine farms, backyard chickens, free-range dairy, equine veterinary hospitals, wildlife rehabilitation centers) and *Salmonella* serogroups (B, C1, C2–C3, D, E1, ND). The backyard chicken category and serogroup B were used as reference variables due to these groups being the most prevalent within our sample. Second, a parallel logistic model was performed using the same variables but restricted to only *Salmonella*-resistant and non-resistant strains. This sub-analysis was employed to better explore whether independent variables affect the prevalence of AMR amongst *Salmonella* isolates. All statistical analyses were performed on RStudio software (http://www.R-project.org, accessed on 10 December 2020) version 3.5.3. Our final model, which included all explanatory variables, was selected based on the best goodness-of-fit according to the Akaike Information Criterion (AIC = 116), compared to models including only one variable at a time (AIC > 130). The statistical significance of each explanatory variable was assessed using Wald’s test and utilizing a 95% confidence level (variable statistically significant if *p*-value < 0.10).

We did not incorporate other characteristics, such as system type (open or closed) and sampling type (environment and animal), due to multicollinearity problems with the variable of system category. Nevertheless, we carried out an exploratory analysis by dropping the latter variables and adding the new set of covariates (system and sampling types).

## 3. Results

### 3.1. Salmonella Presence and Serogroups

In this study, 4047 samples from different animals were analyzed for *Salmonella* presence; 2.61% (*n* = 106) tested positive for *Salmonella* (Figure 1). We predicted the serogroup using the molecular scheme, but in 22 strains, the serogroup was classified as not determined (ND) (Figure 1 and Appendix A). The most common serogroup was the B-type with 48 (45.3%) strains found in all animals that tested positive (Figure 1). *Salmonella* serogroup D (14.1%) was observed in 15 strains from wild animals in rehabilitation centers, wild birds in wetlands and backyard flocks. Similarly, *Salmonella* serogroup C1 (14.1%) was found in 15 strains from an equine hospital, wild animals in rehabilitation centers and backyard chickens. *Salmonella* serogroups C2–C3 (3.8%) were detected in four strains, while serogroup E1 (1.9%) was observed in two strains. Twenty-two strains (20.7%) isolated from wildlife were classified as not determined (Figure 1 and Appendix A).

### 3.2. Antimicrobial Resistance and Susceptibility

A total of 47 out of 106 (44.3% of total strains) *Salmonella* strains analyzed had AMR (*n* = 47). Only 28 of those were SDR, while 19 were MDR. The MDR strains were more frequently observed in closed systems (*n* = 13) compared to free-range (*n* = 6) (Figure 1). Strains with AMR were obtained mostly from closed systems (*n* = 34). These comprised industrial swine (*n* = 8), equine veterinary hospitals (*n* = 20) and wildlife in rehabilitation centers (*n* = 6). On the other hand, AMR in free-range animals was also observed (*n* = 13): free-range dairy farms (*n* = 1), wetland birds (*n* = 1) and backyard chickens (*n* = 11), as shown in Figure 1. In general, the antimicrobials showing the highest levels of resistance were ampicillin (*n* = 27), streptomycin (*n* = 26) and tetracycline (*n* = 26) (see Figure 2). MDR strains were resistant to nine antimicrobials: ampicillin, amoxicillin/clavulanic acid, chloramphenicol, ciprofloxacin, streptomycin, gentamicin, kanamycin, trimethoprim/sulfamethoxazole and tetracycline (Figure 2).

Clustering of the antimicrobial resistance profiles identified in *Salmonella* strains was obtained using a conglomerate-hierarchical with a maximum Euclidean distance of 6.65 points between clusters. Our analysis identified arbitrary cut-off points within the distance spectrum. These specifications were employed to allow the largest number of clusters to be computed (3.32) within two groups of antimicrobial resistance profiles (Figure 2), G1 and G2, which were located at approximately 4.7 distance points apart. G1 included, principally, MDR strains (clusters 2 to 3), while G2 included SDR strains (clusters 4 to 8). G1 grouped 10 strains belonging to closed systems, and all of them were either classified as serogroup B or not determined (ND). In this group, the most extensive antimicrobial-resistance profile was reported as an ampicillin, amoxicillin/clavulanic acid, chloramphenicol, streptomycin, gentamicin, kanamycin, trimethoprim/sulfamethoxazole and tetracycline-resistant strain. On the other hand, cluster 1 grouped two different strains from wildlife in rehabilitation centers with extensive antimicrobial-resistance profiles. G2 included 30 resistant strains from closed and free-range systems classified within serogroups E, ND, B and C. These strains had a profile of limited resistance from one to two antibiotics. The most common resistance profile was triple-resistant to ampicillin, streptomycin and tetracycline (Figure 2).

### 3.3. Likelihood of Antimicrobial-Resistance and Multidrug-Resistance

Table 2 displays the descriptive statistics of our analytical sample. There is a high prevalence of the B serogroup, and most samples were collected from animals (not environmental) in both regression model samples. Diversely, the first model sample presents a large proportion of backyard chicken farms, whereas the group size is smaller in the sample used for Model 2. Wildlife staying at rehabilitation centers displayed the highest proportion in Model 2, and most animals came from closed spaces. Differently, most animals came from open sites in the Model 1 sample (Table 2).

The results of our logistic regression are found in Figure 3. Model 1 shows that *Salmonella* strains from the equine hospital were 2.6 times more likely (odds ratio (OR) = 2.63; 95% CI = 0.95–7.25; *p*-value = 0.061) to be resistant to at least one of the antimicrobials tested, compared to backyard chicken farms (Appendix A). Similarly, our results demonstrated that within industrial swine systems, *Salmonella* strains were 5.08 times more likely to become resistant than backyard chickens (OR = 5.08; 95% CI = 2.09–12.35; *p*-value ≤ 0.001). Moreover, *Salmonella* strains from serogroup D had 0.95 times lower probability of being resistant than the most common serogroup B (OR = 0.05; 95% CI = 0.01–0.47; *p*-value = 0.009). Likewise, serogroup C1 had a lower likelihood of becoming resistant compared to serogroup B (OR = 0.21; 95% CI = 0.05–0.84; *p*-value = 0.027). Industrial dairy farms were not considered due to a lack of variability over AMR prevalence and to avoid perfect multicollinearity.

On the other hand, Model 2 shows the same model after reducing the sample to *Salmonella* (+) strains. Industrial swine isolates were omitted, in contrast with Model 1, as this group did not display variation of the prevalence of AMR, showing perfect collinearity with our dependent variable (Figure 3a). Wildlife in rehabilitation centers and wetland birds had lower odds of acquiring resistant strains compared to backyard chicken farms (OR = 0.16, 95% CI = 0.03–0.89, *p*-value = 0.036; OR = 0.12, 95% CI = 0.01–1.123, *p*-value = 0.075, respectively). Regarding serogroups, our results were consistent with those observed in Model 1. Figure 3b compares both models from Figure 3a by plotting their respective ORs. There are no meaningful differences for the observed characteristics, except for wildlife in rehabilitation centers and the ND serogroup. Figure 3a depicts the average predicted prevalence of AMR by each model. On average, the presence of *Salmonella* with AMR in the samples collected from closed environments was threefold that of their open counterparts (for both models). On the contrary, *Salmonella* with AMR from animal samples collected in open spaces was twice that of closed sites for Model 2 (Appendix A).

Our exploratory analysis partly supports these findings (Appendix A). We included system type (open or closed), sampling type (environment or animal) and serotype group in Model 3 as independent variables to explain AMR (see Table 2). There is a link between samples collected at closed sites and the prevalence of AMR (OR = 3.90; 95% CI = 1.36–11.20; *p*-value = 0.011). No other association was found between the sampling type (collected directly from animals or fecal samples in the environment) and resistance levels (Appendix A). We could not employ further analyses with MDR *Salmonella* due to a lack of variability and multicollinearity problems. However, the hierarchical analysis identified that most MDR profiles (grouped in G1 clusters 2 to 3) were different compared to the profiles grouped in clusters 4 to 8 (see Figure 2 and Figure 3), and cluster 1 was set aside with the highest resistance profile (see Figure 2).

## 4. Discussion

Our study reports 47 antimicrobial-resistant *Salmonella* strains (44.3%). The presence of *Salmonella* strains was mainly observed in serogroups B and C1 within different animal populations in Chile. Coincident with our study, other studies carried out in Chile have found these common serovars to be Enteritidis, Typhimurium and Infantis (serogroup D, B and C1) [15,18,22,24,27,28,29,30]. However, *S.* Enteritidis has represented the highest percentage of isolates [15,16,17,18,19,22]. In Chile, the Institute of Public Health has reported that *S.* Enteritidis is the most common serovar associated with human illnesses [27,28], accounting for 61% of all *Salmonella* cases [27]. In 2019, a number of cases of bacterial gastroenteritis caused by *Salmonella* spp. were reported in Chile [29]. Out of the total 212 outbreaks that occurred over time, 106 originated from *Salmonella* spp. [29]. Similarly to this report, a previous article [30] also indicated a high contamination by *S.* Infantis of chicken meat available for consumption in supermarkets. However, a low prevalence is reported in backyard poultry farms (approximately 1%) [15]. It could be hypothesized that the low isolation is due to the difficulty of recovering and identifying *Salmonella* spp., which further complicates our understanding of the risk of transmission and dissemination among animal and human populations [15,22]. The risk could be greater if these *Salmonella* strains have AMR and even more substantial if MDR strains are presented. Unfortunately, we could not obtain samples from poultry farms due to the strong biosecurity measures and the limited interaction between industry and academia, which is reflected in the lack of studies on these production systems. Yet, studies are needed to achieve a better understanding of the dissemination of *Salmonella* AMR. Therefore, as it is considered relevant, we set about exploring and describing the circulation of *Salmonella* AMR and MDR strains in animal systems in Chile.

Regarding antimicrobial testing, the most common resistance profile was found to be triple-resistant to the ampicillin, streptomycin and tetracycline strains. This is novel because they do not correspond to the frequent treatment received by domestic animals [31]. However, the amount of tetracycline used in marine-farmed salmon 0.7–1.5 (L/kg) corresponds to one of the highest historic values reported by the National Fisheries Service (SERNAPESCA) in Chile [32].

Our results also suggest that *S.* Enteritidis is highly persistent in animal production systems. We reported that serogroup D strains were 18 times less likely to be resistant to at least one antimicrobial agent than the most common serogroup (serogroup B). Our results suggest that serogroups B and C1, including *Salmonella* serovars Typhimurium and Infantis, may have more significant facilities to acquire resistant genes [3]. Our results resemble those found by NARMS and European Union (EU) reports, which have stated that *S.* Enteritidis (serogroup D) had lower antimicrobial-resistance levels than other serovars [10,12]. Coincidentally, we observed through our clustering analysis that MDR strains were grouped in G1, which corresponded to serogroup B and closed systems (horse strains from equine veterinary hospitals and free-range cows). Moreover, cluster 1 grouped two strains from wildlife in rehabilitation centers with broad resistance profiles. In this sense, it has been previously demonstrated that wildlife thriving in anthropized landscapes, such as owls [16] and foxes [33] together with migratory birds [19], may constitute environmental reservoirs of AMR. Consequently, this might explain the acquisition of international clones of BLEA-producing E. coli and S. Infantis (CTX-M) with a broad resistome and virulome. Both strains have been previously detected in Chile and South America [16]. Wild species kept at this human-wild animal interface should be monitored with special attention in the future. Besides, the role of closed production systems—especially in meat production, which is affected by *Campylobacter jejuni* resistance [34], and marine-farmed salmon, which are affected by *Piscirickettsia salmonis* resistance [35]—should also be considered to control the dissemination of AMR and MDR strains in the environment. Examples of contamination may include the use of watercourses risking vegetable uptake and animal consumption [14].

In our results, we also reported that swine production systems were five times more likely to be resistant to at least one antimicrobial (OR = 5.08). This specific result coincides with the study employed by Vico et al. [36], who reported a high prevalence of non-typhoidal *Salmonella* (41.5%; 95% CI: 37.6–45.6%) and a further 86% of MDR strains. Another interesting result was obtained from equine veterinary hospitals, which were 2.63 times more likely to be resistant to at least one antimicrobial [37]. Potential stressful conditions and extensive use of antimicrobials are known to act as trigger factors for AMR in hospitalized animals [37]. To address the former, stewardship programs have been implemented in several countries, such as Switzerland, to avoid further antimicrobial damage to hospitalized animals [38].

Rather than an accurate estimate of the prevalence of AMR in different *Salmonella* serovars, our study can be seen as a point of surveillance of *Salmonella* AMR in a highly unequal country that plays an essential role in global food production [39]. Future research using systematic and more extensive sampling techniques in different animals’ settings could help to more accurately estimate AMR prevalence across different closed systems. This process could help to understand the drivers of AMR and MDR *Salmonella* across farms and within country regions. Data on antimicrobial susceptibility, mainly taken from hospitalized domestic animals, are very limited in numerous developing and highly unequal countries such as Chile [40]. The use of antimicrobials in industrial food production is a significant concern affecting countries worldwide [39]. For instance, the amount and frequency of antimicrobials used on wild animals in captivity has remained mostly unknown, even though a recent study has shed some light on the prevalence of AMR [41]. In addition, salmonellosis surveillance in food, animals and environments has been poorly reported in the literature, with barely any focus on developing and highly unequal countries [42]. It is essential to estimate the risks of AMR to humans, which has remained mostly as a future, but not present, challenge [42]. A recent publication reported an increased prevalence of AMR in zoonotic pathogens, such as *Salmonella* and *Escherichia coli*, in several developing countries [16]. In that study, Chile was classified as an AMR hotspot. Furthermore, the study highlighted the critical nature of delivering point prevalence surveys to enhance the data quality of AMR trajectory modelling [16].

This study has some shortcomings. First, the low number of samples in each system category produced extensive 95% CIs. Second, there was a lack of variability across independent variables, which can be attributed to selection bias. However, this is one of the first studies looking at the presence of *Salmonella* AMR in different animals in Chile by employing a massive testing scale. Third, there was a significant number of backyard chicken farms, which can lead to sampling inconsistencies. Nonetheless, flocks of chicken are one of Chile’s largest animal populations due to the country’s the high demand for chicken meat. Fourth, the numbers of equine and wild animals were relatively high compared to their specific herd populations in Chile. Fifth, multidrug resistance lacked variability; thus, we could not employ further regression analyses for MDR *Salmonella*. Future research is necessary to refine each category’s influence by using a more balanced and greater sample size.

In summary, a better-standardized format to upload and generate data on *Salmonella* AMR could facilitate future estimations of its burden on low-resourced and/or the most impoverished countries [43]. The present study’s results help to improve the existing mechanisms for collecting data on AMR within the region. Moreover, this study evidences the factors that are highly associated with *Salmonella* AMR in a Chilean subsample of wild animals. Stewardship schemes and a well-guided national program aiming to reduce *Salmonella* AMR levels in wild and domesticated animals are essential to help contain further transmission. This study has implications for other similar countries in the Latin American context with similar environments and characteristics, such as Argentina, Brazil, Colombia, Mexico and Venezuela [44].

## 5. Conclusions

This study describes *Salmonella* isolates obtained in different animal systems in Chile by characterizing their serogroups and antimicrobial resistance levels. We report a significant presence of *Salmonella* AMR in animal systems. Among them, the serogroups B and C1 were more frequently observed in *Salmonella* AMR and MDR *Salmonella*. *S.* Enteritidis (serogroup D) had lower AMR levels than other serovars, such as serogroup B. The factors identified in this study could be used for the design of public policies that aim to tackle AMR in the animal industry using a One Health perspective.

## Figures and Tables

**Figure 1 animals-11-01532-f001:**
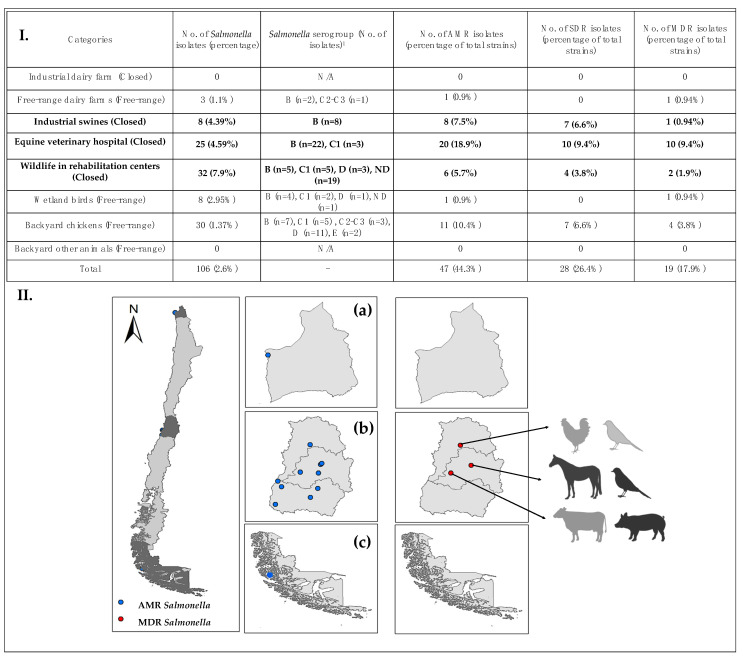
(**I**) Study sites of *Salmonella enterica* isolation from each animal system by categories. Animals from closed systems are shown in black and free-range systems in gray. ^1^ Serogroups were predicted using a molecular scheme. ND: Not determined. AMR stands for antimicrobial-resistance. SDR is AMR to at least one antimicrobial, and MDR means multi-resistant (to three or more antimicrobials). (**II**) Risk maps were constructed using ArcGIS 10 software (Esri, Redlands, CA, USA). (**a**) Northern sites included Region Arica y Parinacota; (**b**) Central zone is represented by Region de Valparaíso, Region Metropolitana and Region Libertador General Bernardo O`Higgins; (**c**) Southern sites are represented by Region Magallanes y Antarctica chilena. Blue circles represent the isolation sites for AMR, while red circles indicate MDR strains.

**Figure 2 animals-11-01532-f002:**
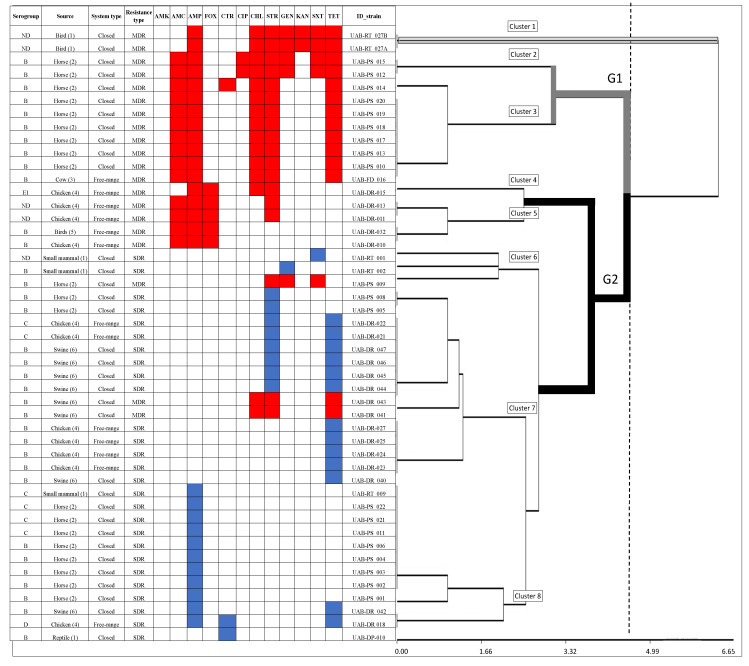
Dendrogram showing clustering of antimicrobial resistance profiles identified in *Salmonella* strains. Dendrogram was obtained from a conglomerate analysis (Infostat version 2017) without restricting the number of clusters. The animal sources where samples were collected corresponded to: (1) wildlife in rehabilitation centers; (2) equine veterinary hospital; (3) free-range dairy farms; (4) chickens backyard farms; (5) wetland birds; (6) industrial swine. Two principal groups were obtained: G1 included MDR strain clusters 2–3, G2 included clusters 4–8 and cluster 1 was set apart with a broad resistance profile. MDR in blue and SDR in red indicate resistance, while white indicates susceptibility. Euclidean distance fluctuation between 0.0–6.5 was marked vertically with a flashing line. Abbreviations: AMC, amoxicillin-clavulanic acid; AMP, ampicillin; FOX, cefoxitin; CTR, ceftriaxone; CIP, ciprofloxacin; CHL, chloramphenicol; STR, streptomycin; GEN, gentamycin; KAN, kanamycin; SXT, trimethoprim/sulfamethoxazole; TE, tetracycline.

**Figure 3 animals-11-01532-f003:**
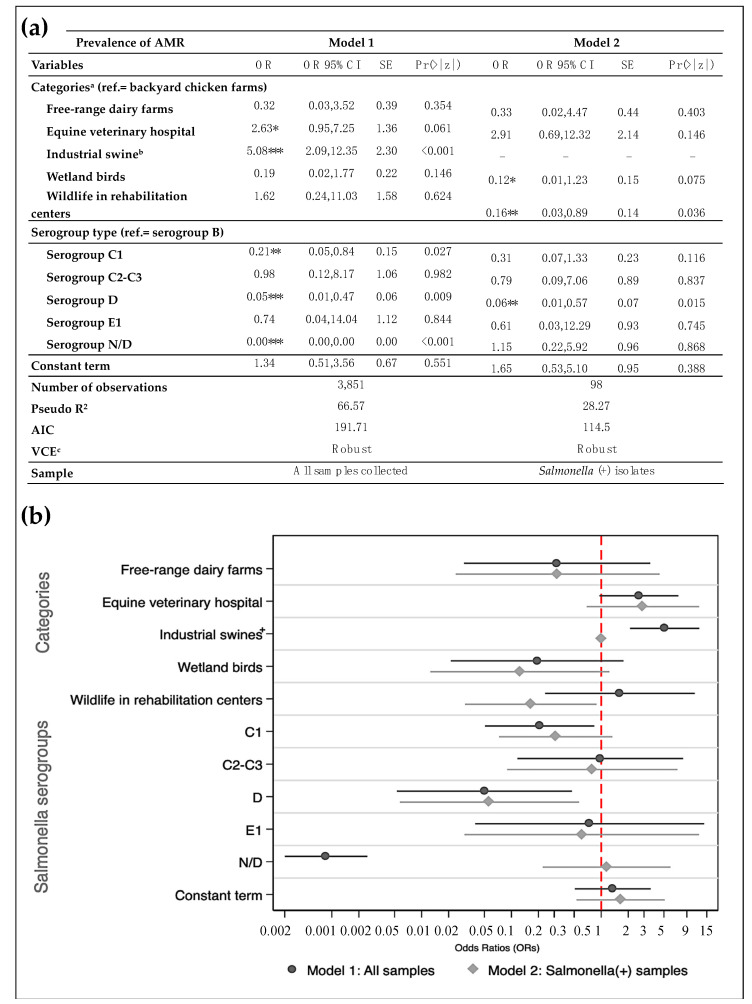
Multivariate logistic regression results, in (**a**) table of results and (**b**) graphical visualization of the model results. Reference group (ref.) stands for backyard chicken farms and serogroup B. Our dependent variable indicates the presence of resistance to at least one antimicrobial for *Salmonella*. SE stands for standard error, whereas OR is for odds ratios. Pr > |z| is for *p*-value; ^a^ Some categories were dropped as they did not present variation of SDR prevalence (i.e., industrial dairy farms and other backyard animals); ^b^ Category dropped in Model 2 due to lack of variation of AMR prevalence. ^c^ Robust standard errors were estimated. * *p* < 0.1, ** *p* < 0.05, *** *p* < 0.01.

**Table 1 animals-11-01532-t001:** Characteristics of the study sites where samples were collected.

Categories	Animal Source(s)	Type	Geographical Location	Sample Type	Isolation Years	No. of Farms/Sites	No. of Samples
Industrial dairy farms	Cows	Closed	Southern Chile	Environmental	2015–2017	8	160
Free-range dairy farms	Cows	Free-range	Central Chile	Environmental	2016–2017	13	260
Industrial swine	Pigs	Closed	Central Chile	Environmental	2015	10	182
Equine veterinary hospitals	Horses	Closed	Central Chile	Environmental and animal	2016–2017	1	545
Wildlife in rehabilitation centers	Small mammals, reptiles and birds	Closed	Central Chile	Animal	2015–2017	3	405
Wetland birds	Birds	Free-range	Northern and Southern Chile	Animal	2016	4	271
Backyard chickens	Chickens	Free-range	Central Chile	Environmental and animal	2013–2015	329	2188
Backyard other animals	Sheep and llamas	Free-range	Northern Chile	Animal	2016	5	36
**Total**							4047

**Table 2 animals-11-01532-t002:** Descriptive statistics by sample used.

Variables	Model 1: Sample		Model 2: Sample	
	MEAN	95% CI	MEAN	95% CI
*Salmonella* prevalence with AMR (%)	1.22	0.87; 1.56	39.8	29.93; 49.66
MDR *Salmonella* prevalence (%)	0.004	0.23; 0.65	16.33	8.88; 23.77
**System**				
Backyard chicken farms (%)	56.82	55.25; 58.38	30.61	21.32; 39.90
Backyard dairy farms (%)	6.75	5.95; 7.54	3.06	−0.41; 6.53
Equine veterinary hospitals (%)	14.15	13.05; 15.25	25.51	16.73; 34.29
Swine farms (%)	4.73	4.05; 5.39	-	-
Wetland birds (%)	7.04	6.22; 7.84	8.16	2.65; 13.68
Wildlife in rehabilitation centers (%)	10.52	6.23; 11.49	32.65	23.21; 41.10
**System type**				
Closed (%)	29.39	27.96; 30.83	58.16	48.22; 68.10
Open (%)	70.61	69.16; 72.00	41.84	31.90; 51.78
**Sampling type**				
Environment (%)	31.24	29.77; 32.70	36.73	27.02;46.45
Animal (%)	68.76	67.30; 70.23	63.27	53.55; 72.98
**Serogroups**				
B (%)	1.25	0.90; 1.60	40.82	30.91; 50.72
C1 (%)	0.39	0.19; 0.59	15.31	8.05; 22.56
C2–C3 (%)	0.10	0.00; 0.21	4.08	0.09; 8.07
D (%)	0.39	0.19; 0.59	15.31	8.05; 22.56
E1 (%)	0.05	−0.02; 0.12	2.04	0.81; 4.89
ND (%)	97.82	97.36; 98.28	22.45	14.04; 30.86
Number of observations	3851		98	
Sample	All samples collected		*Salmonella* (+) strains	

Notes: SD stands for standard deviation and CI for confidence interval.

## Data Availability

The authors of this study assure that the data shared are in accordance with the consent provided by the participants on the use of confidential data.

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
