# Peer review of "Screening the Presence of Non-Typhoidal Salmonella in Different Animal Systems and the Assessment of Antimicrobial Resistance"

_animals, 2021, doi:10.3390/ani11061532_

Round 1

Reviewer 1 Report

Interesting and complete paper. It gives additional information about prevalence of Salmonella and multidrug resistant strains in Chile. The statistical analysis followed seems to me meritorious and correctly applied. I would like to congratulate the authors to publish a study with such an important number of samples tested and highly representative of different environments and animals. Along the paper I made some comments, and questions, maybe by ignorance, to improve the understanding of the paper.

Line 21: please introduce the acronyms SDR, MDR the first time you name them

Line 130-132: this statement seems to be reapeated

Line 176: Which variables were considered?

Line 196: Samonella strains including resistant and not resistant? am I right?

Line 219: 20 % of unidentification seems to be quite high? Is there any explantion for this result? Could you use biochemical identification in those cases?

Line 248: according to the dendogram cluster, clusters 2 and 3 belong to G2 group. I recommend clarifying this point.

Line: 268: What the authors mean with model 1 and model 2? Are they referring to the regression models? I recommend clarifying this point.

Line 309: Which variables and results were included in model 3?

Line 340: Which is NAMRS?

Line 368: it is stimulating how the authors reflect about their own results and the strategies for future experimentation designs.

Figure 3: Check the labels (A) and (B), should not be a) and b)?

Author Response

Dear

Prof. Dr. Clive J. C. Phillips (Editor in chief)

Dr. Javiera Cornejo Kelly (Guest Editor)

Dr. Christopher Hamilton-West (Guest Editor)

We would like to thank you, and the reviewers for these thorough and constructive comments on our paper, and for the opportunity to revise and resubmit to Animals. We have considered and worked through these comments with great attention and dedication including language. The manuscript and supplementary material were attached again but highlighting the main modifications using track changes. We believe our paper is now considerably stronger based on this work. We outline a point-by-point explanation to the provided comments below.

REVIEWER 1 Comments

Interesting and complete paper. It gives additional information about prevalence of Salmonella and multidrug resistant strains in Chile. The statistical analysis followed seems to me meritorious and correctly applied. I would like to congratulate the authors to publish a study with such an important number of samples tested and highly representative of different environments and animals. Along the paper I made some comments, and questions, maybe by ignorance, to improve the understanding of the paper.

  • Line 21: please introduce the acronyms SDR, MDR the first time you name them

Answer: The acronyms have been modified as suggested

  • Line 130-132: this statement seems to be repeated

Answer: We have shortened the paragraph and removed repeated information

  • Line 176: Which variables were considered?

Answer: We have stated which independent variables were incorporated as independent variables now. We included serogroup type (B, C1, C2-C3, D, E1, N/D) and animal category (industrial swine, wetland birds, equine veterinary animal, free-range dairy animals and backyard chicken).

  • Line 196: Samonella strains including resistant and not resistant? am I right?

Answer: We have clarified what you have pointed out in the main manuscript now. We have included Salmonella resistant and not-resistant strains.

  • Line 219: 20 % of unidentification seems to be quite high? Is there any explanation for this result? Could you use biochemical identification in those cases?

Answer: The strains not classified within any serogroup possibly correspond to serogroups that are complex to be identified by the molecular techniques used. In reference to biochemical tests or other laboratory phenotypic analyses, these techniques do not allow the serotyping of Salmonella spp.

  • Line 248: according to the dendogram cluster, clusters 2 and 3 belong to G2 group. I recommend clarifying this point.

Answer: This has been clarified accordingly in line 239.

  • Line: 268: What the authors mean with model 1 and model 2? Are they referring to the regression models? I recommend clarifying this point.

Answer: Yes, we are referring to the regression models. This has been clarified in the manuscript.

  • Line 309: Which variables and results were included in model 3?

Answer: We added the following sentence to explain model 3; We included system type (open or close), sampling type (environment or animal) and serotype group in model 3 as independent variables to explain AMR (see Table 2).

  • Line 340: Which is NAMRS?

Answer: We have spelled the abbreviation out now correctly in the manuscript. It stands for the National Antimicrobial Resistance Monitoring System (NARMS).

  • Line 368: it is stimulating how the authors reflect about their own results and the strategies for future experimentation designs.

Answer: We hope to contribute with this work to the knowledge of resistant Salmonella and their potential for transmission to humans.

  • Figure 3: Check the labels (A) and (B), should not be a) and b)?

Answer: This was now modified accordingly in Figure 3.

Sincerely,

Reviewer 2 Report

Dear Authors

your manuscript  is well structured and is of importance for  Public Health, also considering the poor available literature  on this topic. The applied statistical analyses and the reading of the results show a great deal of work . However, I think that in spite of this amount of work carried out, the discussion could be more exhaustive to optimize the quality of your  manuscript .

Please  find my remarks and suggestions in attachment

Author Response

REVIEWER 2 Comments

your manuscript is well structured and is of importance for Public Health, also considering the poor available literature on this topic. The applied statistical analyses and the reading of the results show a great deal of work. However, I think that in spite of this amount of work carried out, the discussion could be more exhaustive to optimize the quality of your manuscript. Please find my remarks and suggestions in attachment.

  • Line 38 It is a repetitive (check line 34)

Answer: The information considered in each line is different, in line 34” their serogroups were characterized and tested for susceptibility to 16 different antimicrobials” and in line 38 “Of the 47, 28 corresponded to single-drug resistance (SDR=26.4%) and 19 to MDR (17.9%). S. Enteritidis was highly persistent in animal production systems”

Material and Methods

  • It could be interesting for your discussion include samples from chicken industrial farms (closed).

Answer: In Chile, due to the characteristics of industrial poultry production systems, it is difficult to take samples from these systems, mainly because of the strict biosafety standards since they keep animals ranging from broiler hens to broiler chickens. Another relevant factor is the lack of interaction between the poultry industry and academia (university). This is reflected in the lack of studies that have analyzed these systems.

  • Line 140 2.3 Serogroup characterization and antimicrobial susceptibility It is my doubt but it's probably my fault. Once you have detected the serogroup, how did you performed a typing of Salmonella?

Answer: This process involves two PCRs: i) multiplex PCR to identify the main serogroups, followed by ii) a second PCR to identify the flagellar phases through the H1 and H2 genes. Finally, serotyping is obtained by sequencing the PCR products of these H1 and H2 genes based on the study published by Ranieri et al. (2013).

Discussion

  • Line 328 You did not test chicken industrial farm. They could have different situation of backyard. The stocking density, large size and stress can increase occurrence of Salmonella However, a low prevalence is reported in poultry farms (approximately 1%) , so it could be hypothesized that the low isolation is due to the difficulty of recovering and identifying Salmonella spp. which further complicates the understanding of the risk of transmission and dissemination among animal population and human population

Answer: It is expected that the industry will collaborate more closely with academia (Universities) in the future, but as mentioned above, it is highly difficult to enter these systems to take samples to evaluate salmonella strains (a new sentence was added between lines 340-345 and 362-374).

  • I would move sentence line 359 to 389. Short comings (critical points) are more suitable in the final speculations. Consequently line 370 is moved to line 359.

Answer: We changed what you pointed out.

  • Line 335 Interestingly whereas that S. Enteritidis is highly persistent in animal production systems, we reported that serogroup D strains were 18 times less likely to be resistant to at least one antimicrobial agent than the most common serogroup (serogroup B). Our results suggest that serogroup B and C1, including Salmonella serovars; Typhimurium and Infantis, may have more significant facilities to acquire resistance genes (3). Our results resemble those found in NAMRS and the European Union0(EU) reports in which S. Enteritidis (serogroup D) had lower antimicrobial- resist.

If you have not typed (see my question in material and methods) you may consider that can be other Salmonella, i.e. S. Dublin, as they are included in serogroup D. In this case it is not really surprising that S.Infantis and S.Typhimurium can have more probability to be resistant.

Answer: Indeed, it is a probability, but lower than S. Enteritidis. This is demonstrated by all the work carried out in the last time in Chile and the official information produced by the sanitary department. However, it would be interesting to consider serotyping all isolates in future studies.

  • Finally, several data, although numerous obtained on the basis of a modest number of samples (a critical point correctly specified by you) could led to interesting hypotheses. Ex: low resistance to ciprofloxacin and high resistance to tetracycline although they are 2 widely used antibiotics. The presence of resistance in the wild animal of rehabilitation centers, or free ranged animals or wet land birds could be derived from the external environment contaminated by resistant bacteria. Furthermore, beyond the extensive use of the antibiotic in animals, the vertical transmission of resistance genes should be taken into consideration.

Answer: We have added information in lines 346-351 regarding this comment (see manuscript)

Table and figures

  • Swines vs swine

Answer: We have changed the words accordingly throughout the manuscript.

  • 261 groups vs group

Answer: We have changed the word to its plural form.

  • Figure 3 Line 303 Please check quotation (A) You specify or and Se but not Pr (z)

Answer: We have clarified this now under Figure 3 within the notes.

Sincerely,
